# Organs-On-Chip Models of the Female Reproductive System

**DOI:** 10.3390/bioengineering6040103

**Published:** 2019-11-07

**Authors:** Vanessa Mancini, Virginia Pensabene

**Affiliations:** 1School of Electronic and Electrical Engineering, University of Leeds, Leeds LS2 9JT, UK; elvm@leeds.ac.uk; 2School of Medicine, University of Leeds, Leeds LS2 9JT, UK

**Keywords:** reproductive organs, women health, preterm

## Abstract

Microfluidic-based technology attracts great interest in cell biology and medicine, in virtue of the ability to better mimic the in vivo cell microenvironment compared to conventional macroscale cell culture platforms. Recent Organs-on-chip (OoC) models allow to reproduce in vitro tissue and organ-level functions of living organs and systems. These models have been applied for the study of specific functions of the female reproductive tract, which is composed of several organs interconnected through intricate endocrine pathways and communication mechanisms. To date, a disease and toxicology study of this system has been difficult to perform. Thus, there is a compelling need to develop innovative platforms for the generation of disease model and for performing drug toxicity/screening in vitro studies. This review is focused on the analysis of recently published OoC models that recreate pathological and physiological characteristics of the female reproductive organs and tissues. These models aim to be used to assess changes in metabolic activity of the specific cell types and the effect of exposure to hormonal treatment or chemical substances on some aspects of reproduction and fertility. We examined these models in terms of device specifications, operating procedures, accuracy for studying the biochemical and functional activity of living tissues and the paracrine signalling that occurs within the different tissues. These models represent a powerful tool for understanding important diseases and syndromes affecting women all around the world. Immediate adoption of these models will allow to clarify diseases, causes and adverse events occurring during pregnancy such as pre-eclampsia, infertility or preterm birth, endometriosis and infertility.

## 1. Introduction

Novel three-dimensional (3D) cell culture devices, called “Organs-on-chip” (OoC) allow to reproduce in vitro tissue and organ-level functions of living organs and systems. These models have been applied for the study of specific functions of the female reproductive tract, which is composed of several organs interconnected through intricate endocrine pathways and communication mechanisms. To date, a disease and toxicology study of this system has been difficult to perform [1]. Furthermore, despite there being increasing efforts in the research and development of new pharmaceutical products, reported trends suggest that fewer new drugs have been brought to the market over recent decades [2,3]. Thus, there is a compelling need to develop innovative platforms for the generation of disease model and for performing drug toxicity/screening in vitro studies. 

This review analyses examples of OoC models that allow the co-culture of different cell types and the in vitro study of pathological and physiological conditions of the female reproductive tract and changes occurring during conception and pregnancy. These models aim to be used to assess changes in metabolic activity of the specific cell types and the effect of exposure to hormonal treatment or chemical substances on some aspects of reproduction and fertility. In detail, these innovative OoC models have the potential to facilitate accurate studies of the biochemical and functional activity of living tissues and to evaluate the paracrine signalling that occurs within the tissues. In particular, these models will represent a powerful tool to study complications that may occur during pregnancy such as pre-eclampsia, infertility or preterm birth. Preterm birth (PTB), for example, is caused by spontaneous preterm rupture of the foetal membranes and currently represents the leading cause on neonatal death [4]. Clinical, epidemiologic and experimental studies have linked environmental toxicant exposure [5] and microbial induced infection [6] with a decreased fertility and an increased preterm birth rate. However, practical and ethical barriers limited the possibility to further clarify this correlation in humans, being impossible to recruit pregnant women or collect tissues during gestation. Thus, new OoC models of the uterine environment, placenta and foetal membranes could be employed for assessing the effects of environmental insults, such as infection or inflammation, which may lead to preterm birth.

Finally, human patient-derived cells are used in these OoC, thus these models are not only representative of the human physiology, but they could be used for the definition of personalised therapies for patients affected by diseases, such as endometriosis and infertility. Endometriosis in fact is a chronic disease affecting 10% of women worldwide, approximately 170 million [7]. It is associated with the presence of tissue containing endometrial, epithelial and stroma cells outside uterus. Most studies of human endometriosis rely on the use of animal models, which are expensive and do not reliably mimic human physiological characteristics of endometrium [8]. To this aim, the development of novel in vitro platforms able to mimic the human pathophysiology of peritoneal endometriosis represents a great potential for the analysis of this disease.

## 2. The Female Reproductive System

The female reproductive system is an interconnected series of organs that support the production of steroid hormones, production of oocytes, union of egg and sperm and development of the foetus [9]. The organs of the female reproductive tract work as a cohesive system and are dynamically synchronised in order to guide ovulated oocytes, prepare for implantation and nurture a foetus to develop as an independent organism. The main organs of the female reproductive tract are the ovaries, fallopian tubes, uterus and cervix (Figure 1) [1,10].

The ovary serves two essential functions in female reproduction: development of the female gametes (oocytes) and the synthesis and release of steroid hormones. Each oocyte is enclosed in the ovarian follicle, surrounded by somatic cells, the granulosa and theca cells. The oocyte and the follicular wall are separated by a thin transparent membrane, the zona pellucida, which is secreted by the oocyte.

The fallopian tube has distal opening in the form of finger-like projections, called fimbriae that facilitate uptake of the ovulated oocyte and its transport towards the uterus. 

If an oocyte is fertilised in the fallopian tube, the blastocyst migrates to the uterus where pregnancy is established. The wall of the uterus has three different layers of tissue, which are the perimetrium, which is the external layer, the myometrium and the endometrium, which is the layer of the uterus that lines the uterine cavity. Endometrium is composed of an internal surface layer, the stratum functionalis, consisting of a lining epithelium and uterine glands that is shed during menstruation, and the stratum basalis, which is not shed during menstruation, but contains blood vessels that produce part of the menstrual flow. The neck of the uterus terminates in the cervix that connects the vagina with the uterine cavity and acts as a barrier between the vagina and the rest of the reproductive tract [9]. 

### 2.1. Menstrual Cycle

Each menstrual cycle can be divided into three main phases: the menstrual phase (or menses), the follicular phase and the luteal phase. During each reproductive cycle few primordial follicles, which are formed during foetal life and held in a meiotic arrest, resume growth to produce a mature oocyte. However, most activated primordial follicles recruited each cycle undergo atresia, leaving a singular dominant follicle to reach ovulation [1].

Follicular phase is characterised by the rapid growth of ovarian follicles promoted by follicle stimulating hormone (FSH) from the pituitary gland. Theca cells produce androstenedione, which is converted into oestrogen by adjacent granulosa cells [11]. The rise in oestrogen causes growth of the endometrium and enlargement of uterine glands. Levels of oestradiol in the blood continue to rise throughout most of the follicular phase and initiate a negative feedback loop that represses FSH release from the pituitary. About 24 to 48 h after this peak in oestradiol, a luteinizing hormone (LH) surge from the pituitary occurs in the blood. This LH surge initiates the resumption of meiosis in the oocyte within the largest follicle and causes the release of the mature oocyte into the fallopian tube (ovulation). After ovulation, granulosa cells from the ovulated follicle differentiate to form corpus luteum. The luteal phase lasts from the ovulation to the beginning of menstruation and is characterised by the secretion of oestradiol and progesterone by the corpus luteum. Levels of oestradiol and progesterone rise in the middle of the luteal phase. Progesterone prepares the uterus for the arrival of an embryo, increasing endometrial thickening and glandular secretion of nutrient that will be used by the embryo if one is conceived. This process is called “decidualization” of the endometrial stromal compartment and is essential for the successful establishment and maintenance of pregnancy. It consists in the morphological and biochemical differentiation of the endometrial stromal fibroblasts in preparation for pregnancy [12].

The presence of oestradiol and progesterone in the luteal phase results in negative feedback on both FSH and LH secretion, which are relatively low in the luteal phase. In the absence of a fertilised egg, the corpus luteum begins to degenerate and progesterone levels decline, which initiate endometrial lining degeneration and discharge during menstruation. After menstruation, the stratum basalis gives rise to a new stratum functionalis. In the event of successful fertilization and subsequent implantation, human chorionic gonadotropin (hCG) produced by the blastocyst signals to the ovary to retain the corpus luteum. In turn, progesterone from the ovary preserves the uterine lining to support pregnancy [1,9].

### 2.2. Pregnancy and Foetal Membranes

If fertilization occurs, the fertilised egg (zygote) moves down the oviduct and into the uterus, where it implants in the uterine wall (Figure 1). The zygote undergoes several cleavage divisions to become first a ball of cells (morula) and then a larger mass of cells called blastocyst which enters the uterus. Blastocyst has a fluid-filled cavity (blastocoel) and an outer layer composed of a single layer of cells, the trophoblast, just inside the zona pellucida. A clump of cells near one end of the blastocyst, underneath the trophoblast layer, forms the inner cell mass that will give rise to the embryo. On about the sixth day after fertilization, the uterus secretes an enzyme that dissolved the zona pellucida surrounding the blastocyst and leads the inner cell mass to attach the uterine wall, initiating implantation. During the early phase of implantation, the blastocyst begins to invade the endometrium and the trophoblast differentiates into an outer syncytiotrophoblast and an inner cytotrophoblast. Meanwhile, cells in the uterine stroma (the connective tissue framework of the uterus) multiply rapidly and form a cup that grows over the blastocyst until implantation is complete, called uterine decidua. The syncytiotrophoblast encloses the developing embryo and is the only embryonic tissue in direct contact with maternal tissue throughout pregnancy. It will later form the external layer of the placenta [9]. 

The inner cell mass produces three of the four extraembryonic membranes: the yolk sac, the amnion and the allantois (Figure 2). The yolk sac is and endoderm-lined membrane that surrounds the blastocoel and supplies the embryo with blood cells. It is not functional in humans and degenerates early in the development. The amnion is a membrane composed of a single layer of tight and dark epithelial cells that grows over the forming embryo. The amnionic cavity becomes filled with amnionic fluid that supports and protects the foetus against mechanical shocks and supplies water and other nutrients to the foetus. Moreover, the amnion is expandable and flexible in size as it tries to accommodate the development of embryo to its later stages. Allantois forms as a small pouch at the posterior end of the embryo and briefly extends into the umbilical cord. Finally, the chorion is an extraembryonic membrane which is derived from the cytotrophoblast and surrounds the embryo and the other membranes. While amnion is avascular, chorion is vascular and it contains mononuclear trophoblasts. The chorion eventually fuses with the amnion and is in direct physical contact with uterine decidua, which contains differentiated endometrial stromal cells [9,13].

Human foetal membranes represent the maternal-foetal interface and provide a fundamental protective role for the foetus. In particular, these membranes protect the foetus against mechanical stress and potentially harmful agents, such as bacteria and viruses, during pregnancy. They also serve as a target for regulation by hormones and molecules that are fundamental for the development of foetus and maintenance of pregnancy [14]. 

Term and preterm labour involves a series of events among which the decidual/membrane activation, which implies the separation of the chorioamniotic membranes from the decidua, and eventually the rupture of the membrane. 

## 3. Organs-On-Chip Models of the Female Reproductive Tract

With the great advance in microfluidics and organs-on-chip research, in the last few years several groups have reported the development of 3D microengineered devices for the replication of parts of the female reproductive system (Table 1). These include the modelling of the human placenta [15,16] and the uterus [17]. More recent works demonstrated the possibility to connect different tissues in a complex microfluidic device to reproduce the 28-days menstrual cycle [18]. In the next section, features of these relevant examples are discussed and analysed with a special focus on chip design methods used to culture different human derived cells and endpoints selected for validation.

### 3.1. Placenta-On-a-Chip Models

The placenta is a highly-specialised organ in the human body that plays the critical role of mediating the exchange of various endogenous and exogenous substances between mother and foetus during pregnancy. The transfer of substances such as oxygen, nutrients and foetal metabolic waste is mediated by the “placental barrier”, a multi-layered membranous structure composed of trophoblasts, connective tissue, basal lamina and the foetal endothelium [19]. The regulation and selectivity of this barrier are fundamental in the progression of both placental and foetal development, and critical changes of its function can lead to complications for the pregnancy [20]. Current studies of structure and functions of the placenta mainly rely on animal models [21], which do not reliably mimic the human physiological characteristics of the maternal-foetal interface, or in vitro studies based on transwell cell culture inserts for the culture of trophoblasts that fail to reproduce the multi-layered structure of the placental barrier [22]. 

Blundell et al. have developed an in vitro microengineered cell culture system that mimics the architecture of the human placental barrier by co-culturing human trophoblast cells and human placental villous endothelial cells under dynamic flow conditions. The model is a compartmentalised microfluidic system composed by upper and lower microchannels separated by a thin semipermeable membrane [15]. The upper and lower layers of the microdevice were fabricated in Polydimethylsiloxane (PDMS) using standard soft lithography techniques [23]. Human primary placental villous endothelial cells (HPVECs) isolated from term placentas and BeWo b30 human trophoblast cell line have been used. After sterilization with UV irradiation and extracellular matrix (ECM) coating of the porous membrane, a suspension of trypsinised HPVECs was introduced into the lower microchannel and the device was immediately inverted to allow the cells to adhere to the lower side of the porous membrane. Once cell attachment was established, the device was flipped back, and the upper microchannel was seeded with a suspension of BeWo cells. The structural integrity of the barrier was assessed evaluating the formation of cell-cell junctions which was uniform across the entire porous membrane. In addition, the apical surface of the trophoblasts cultured in the upper chamber for 3 days demonstrated the presence of a dense layer of microvilli, which are typical membrane protrusion that serve as a key regulator of placental transport. This “placental-barrier” model also made it possible to engineer the soluble microenvironment of trophoblast cells in the maternal compartment using forskolin supplemented media to induce their syncytialization. Syncytialization is the process in which cytotrophoblast cells covering the chorionic villi of the human placenta differentiate and fuse to form a multinucleated syncytiotrophoblast and this process can be induced by the activation of protein kinase A by forskolin [24]. The progressive fusion of the cultured trophoblast cells and their increased production of human chorionic gonadotropin (hCG) was observed, indicating successful differentiation of the trophoblast epithelium and the ability of this placenta-on-a-chip platform to enable syncytialization. Finally, the transport of glucose across the placental barrier was quantitatively analysed by creating a concentration gradient across the barrier. The per cent increase in foetal glucose concentration in the co-culture model resulted considerably reduced compared to two other models composed of a bare membrane and a monolayer of BeWo cells without the endothelium, demonstrating the ability of the differentiated trophoblasts to mediate glucose transport with physiological relevant rates.

In another recent work [16], Lee et al. proposed a similar “placenta-on-a-chip” microengineered device to model the organ-specific architecture and physiological microenvironment of the placental barrier that enables the compartmentalised perfusion co-culture of human trophoblasts (JEG-3) and human umbilical vein endothelial cells (HUVECs) on a thin extracellular matrix (ECM) membrane. The placenta-on-a-chip microdevice consists of two PDMS slabs generated using standard soft lithography technique containing microchannel features separated by a vitrified collagen membrane. The upper channel corresponds to the foetal capillary compartment while the lower channel corresponds to the intervillous space. For the formation of the vitrified collagen membrane, a solution of collagen type I was gently dispensed on the central part of the lower microchannel and then incubated at 37 °C for 40 min for gelation. Subsequently, the collagen gel was dried overnight at room temperature to produce a vitrified collagen membrane that remained attached to the lower PDMS microchannel. Finally, the PDMS surfaces of the upper and lower channels were aligned and bonded together using a plasma cleaner to produce a sealed microfluidic device.

After UV irradiation, the upper and lower surfaces of the vitrified membrane were coated with fibronectin and gelatin to promote cell attachment and growth. Then, trophoblasts (JEG-3) cells and green fluorescent protein (GFP)-expressing HUVECs were gently injected and allowed to adhere to the lower and upper sides of the membrane, respectively.

Results demonstrated that microfluidic culture conditions promoted the proliferation and maturation of JEG-3 cells and HUVECs which produced highly confluent monolayers on either side of the membrane. The resultant bi-layer tissue resembled the morphological structure of the placental barrier in vivo. To evaluate the transport of glucose from the maternal to the foetal side in this model, the lower trophoblast culture chamber was perfused with medium containing a higher concentration of glucose, to recapitulate the physiological gradient across the maternal-foetal interface. In the co-culture model the glucose transfer was found to be the smallest, illustrating the ability of the bi-layer microarchitecture to limit molecular transport across the barrier. These results suggest that the transport of glucose across the barrier in this model is regulated not only by concentration gradient-induced passive diffusion, but also by other mechanisms in which cells play a critical role. 

In conclusion, both these compartmentalised three-dimensional microsystems allowed the recapitulation of the in vivo human placental barrier, mimicking its physiological microstructure and transporter functions. These placenta-on-a-chip systems were used for in vitro study of the integrity of the barrier and the molecular transport across the maternal-foetal interface. Both models adopt dynamic perfusion, with flow rates ranging from 30 to 100 L/min. Previous studies have shown in microfluidic devices that the formation of placental microvilli from BeWo trophoblastic cells and villous trophoblasts is mediated not only by intracellular factors (Ca ions, ezrin/radixin/moesin proteins), but also by the extracellular fluid shear stress that allows to establish a tight epithelium that control the maternal/foetal interface and to form the microvillar structures of the placental barrier cells [25].

The integration of a vitrified collagen membrane is beneficial by promoting cell adhesion and growth. Improvements of the systems could include the culture of primary villous trophoblast cells and the investigation of the transport of other nutrients rather than glucose, such as small molecules and biologics, to further validate the physiological relevance of the model. For example, the transport across the barrier of harmful agents such as bacteria, viruses and parasites could be evaluated to study infection transmission from mother to foetus. In addition, different cell types of human placenta (e.g., decidual cells, maternal macrophages, leukocytes) might be included to obtain a more complex model of the whole human maternal-foetal barrier.

Models of healthy placental barrier favour the understanding of important functions of the maternal-foetal interface, such as the placental transfer of exogenous and endogenous substances. By mimicking placental dysfunctions that are known to be related to pathological conditions, such as foetal growth restriction, diabetes and pre-eclampsia, these OoC could enable the development and screening of new drugs and alternative therapeutic approaches

### 3.2. Artificial Uterus on a Microfluidic Chip

In vitro fertilization-embryo transplantation (IVF-ET) is the process of combining egg and sperm outside of female’s body with the aim of achieving fertilization [26]. Although IVF-ET is considered the major treatment for infertility, some limitations are associated with its application, such as low fertilization rate and increased incidence of multifertilization [27]. Over the last decade, some studies have reported the use of microfluidic systems for in vitro fertilization [28] and embryo development [29] with the aim to overcome the limitations of IVF-ET techniques. 

As an example, in a recent study Wei-Xuan et al. developed a microfluidic “uterus-on-a-chip” that can be used for IVF-ET procedures replicating uterine functions such as ovulation, insemination and embryo development [17]. The microfluidic “uterus-on-a chip-chip” contains two PDMS layers separated by a porous polycarbonate (PC) membrane used as support for endometrial cell culture. The top layer is a zigzag shaped channel containing a series of interlaced microsievers that allow the capture of oocytes, while the bottom layer includes four perfusion parallel channels with an array of micropillars on the bed that serves as a support for the porous membrane [30]. 

After device sterilization with UV light and treatment with gelatin solution, endometrial cell suspension was introduced into the microchannel of the top layer to adhere to the porous membrane. After 8 h of perfusion with endometrium medium, oocyte suspension was flushed into the top layer microchannel and captured by the microsievers. After 2 days of co-culture, capacitated sperms were introduced into the microchannel and the formation of the blastocyst was observed.

This microfluidic chip allows the co-culture of oocyte and endometrial cells mimicking the structure and function of the uterus and facilitating molecule diffusion through the membrane. Furthermore, in the OoC, a higher morula and blastocyst rates was observed compared to embryo development in a conventional Petri dish. These results can be explained by the benefits of: (i) perfusion culture, which ensures the sufficient amount of oxygen and nutrient for embryo development and the elimination of metabolic waste and (ii) the porous membrane which enables the interaction of the embryo with the underlying cells and the diffusion of soluble factors, which provided a more appropriate environment for the development of embryos. 

This study represents a powerful alternative to conventional IVF-ET procedure and demonstrates the beneficial aspects of embryo co-culture with endometrial cells, which facilitate in vitro embryo development. This study is limited to mouse oocytes but the method can be potentially relevant for human physiology.

To address limitations of conventional 3D cell culture studies on endometrial biology and pathophysiology, Gnecco et al. recently developed a human “endometrium-on-chip” model to investigate cross-talk between human perivascular stroma and endothelial cells within a microengineered compartmentalised chip [31]. A sex steroid hormonal treatment was used to recapitulate physiological changes which occur during the reproductive cycle in vivo. Stromal decidualization was observed after 10 days of culture by measuring increased level of prolactin from the collected media and by noticing typical morphological changes of cell shape. Moreover, perfusion was applied to recreate shear stresses able to induce endothelial cell polarization that was successfully observed by fluorescence microscopy. The innovation and potential of this endometrial cell culture platform for testing pharmaceutical or toxicant agents might be further improved by the integration of additional cell types to recreate a more complex model of the human endometrium. 

In the field of fertility and reproduction, a recent study worth mentioning is the one conducted by Nagashima et al. describing a dynamic in vitro culture of cat and dog ovarian follicles implemented within a microfluidic chip. Follicles enclosed within the ovarian cortex as well as those isolated from ovarian cortex were cultured in the chip under various flow rates and compared with control agarose gel cultures. Microfluidics allowed to overcome limitations of conventional ovarian tissue culture enabling the monitoring of follicle development over time and the continuous nutrient replenishment. However, no significant differences in terms of follicle morphology and growth were observed between the microfluidic and the control cultures and also among the different tested flow rates [32]. 

These models, combined with recently developed in vitro protocols for the generation tumour organoids [33], could represent a powerful system for modelling patient-specific endometrial and uterine cancer and to identify personalised treatments.

### 3.3. In Vitro Model of Endometriosis

Microengineered devices allow the in vitro modelling of cell-cell interaction; thus, they can be used for understanding the pathogenesis of diseases such as tumour and endometriosis. Endometriosis is a chronic disease associated with the presence of tissue containing endometrial, epithelial and stroma cells outside uterus. Most studies of human endometriosis rely on the use of animal models, which present several drawbacks such as the lack of menstrual cycle in rodent, mice and rabbit models, and the rarity and cost of macaques or baboons. To this aim, the development of novel in vitro platform able to mimic the human pathophysiology of peritoneal endometriosis represents a great potential for the analysis of this disease. 

Chen et al. presented an in vitro model for the dynamic real-time monitoring of the microenvironmental interactions in endometriosis [8]. In detail, they developed a microfluidic system where endometrial stromal cells (ESCs) and human peritoneal mesothelial cells (HPMCs) are at first patterned in separate areas to simulate their growth in vitro and then co-cultured in a common compartment to “visualise” mutual interactions.

ESCs were isolated from human endometrium, while HPMCs were isolated from peritoneum collected from the lateral abdominal wall, which is a site usually not prone to endometriosis. 

The experiments consist in the measurement of the migration speeds of these two cell types in the microfabricated device and were carried out with different cell types obtained from endometriotic (em) individuals (ESCs em and HPMCs em) and from control (con) individuals (ESCs con and HPMCs con). It was observed that HPMCs from control individuals have the ability to resist the invasion of ESCs and expel the growth of ESCs either from endometriotic or control individuals. However, HPMCs from endometriotic individuals cannot effectively resist the invasion of ESCs. When ESCs con or ESCs em contacted and interacted with HPMCs em, endometrial stromal cells invaded into areas of HPMCs em and led to their disappearance. In particular, HPMCs lost contacts between each other, detached from the substrate, and disappeared under invasion of ESCs em. These results suggest that in vivo ESCs are not able to adhere on healthy peritoneum and endometriosis is probably related not only to endometrial conditions, but also to the pathogenic sites. They found that not only pathological HPMCs lost their protective role in resisting invasion of ESCs, but they also might have a basic pathological role in the generation and development of endometriosis.

In conclusion, this platform enabled the study of pathophysiological conditions of endometriosis and can be suitable for further investigation of this disease or other biological processes through the modelling of the interaction between different cell types. The design of the device limits its application to studies where cells are able to migrate freely and directly interact with other cell types, whereas in many in vitro studies, the use of a porous membrane is still required for the separation of the different cell population. However, a positive aspect of this study is the use of primary human cells, which fosters the validity of the system as a relevant substitute of the in vivo model.

### 3.4. Modelling the Human Reproductive Tract and the Menstrual Cycle

In 2013, Woodruff’s team created the first “organ-on-a-chip” model that functionally re-creates in vitro the entire 28-days menstrual cycle, where human and mouse cells from several reproductive organs were grown in a network of tiny and interconnected microengineered units [18]. Three different microfluidic platforms (MFP) termed Solo-MFP, Duet-MFP and Quintet-MFP were developed and used as a platform able to sustain tissue-level functions for the length of the human menstrual cycle. The Solo-MFP and Duet-MFP systems were designed for single tissue culture and are based on pneumatic actuation technology that uses sequential application of pressure and vacuum to valve and pump membranes to enable fluid to move throughout the system. The Quintet-MFP was designed for multiple tissue culture and is based on embedded electromagnetic actuation technology which uses a series of micropumps and individually controllable actuators that enable precise flow control. Thus, the use of tubes, valves and pumps allows to push air and fluids through the system, mimicking the body’s natural circulation.

Firstly, mouse ovarian tissue was cultured in the Solo-MFP system for 28 days, to test whether the microfluidic system supports follicle growth. Follicle-stimulating hormone (FSH) was provided through the first 14 days (day 14 to day 0) to mimic follicular phase gonadotropin levels. In addition, to phenocopy the luteinizing hormone (LH) surge, they created an algorithm called ‘surge-purge’ that generated peak human chorionic gonadotropin (hCG) on day 0. Hormones were then brought to baseline for the remaining days of culture (day 1 to day 14). Under the effect of hormone treatment, this microfluidic environment demonstrated to be capable of supporting individual ovarian follicle growth, maturation, ovulation, and differentiation of granulosa cell into luteal cells. 

Successively, follicular hormone secretion (oestradiol E2, progesterone P4, inhibin A and inhibin B) was investigated and results showed that, compared to static culture, follicles exposed to dynamic flow had significantly higher E2 and P4 production. Similar results were obtained culturing ovarian explants collected from CD-1 mice in either the Solo-MFP or Duet-MFP for 28 days.

On the other hand, the Quintet-MFP system comprises five “organs” linked together by a blood-like liquid carrying hormones, cell signalling molecules and drugs. In particular, fallopian tubes, uterus, ectocervix and liver were obtained from human tissues collected from women undergoing hysterectomies, while ovaries were obtained from mouse tissue, because healthy ovaries are rarely removed from women. However, mouse ovarian cells produce the same hormones as human ovaries; thus, they represent a reliable model. Tissues were cultured for 28 days and exposed to the pituitary hormone circulation described above. In response to hormone treatment, the cells secreted levels of oestrogen and progesterone found in a typical menstrual cycle replicating the signalling that occurs among different female reproductive organs. In addition, this microfluidic system was used to mimic the ‘pregnancy’-like state, prolonging the luteal phase function and simulating hormone activity that takes place after implantation. In particular, the corpus luteum was maintained for the full 14 days of the luteal phase following ovulation and the “pregnant” luteal tissue demonstrated able to produce significantly higher levels of progesterone P4 compared to the “non-pregnant” system. Moreover, a liver microtissue was integrated in the system because it can eventually metabolise drugs tested using this platform. 

This innovative system allowed the modelling of multi-tissues interaction and the reproduction of characteristic reproductive functions, providing another avenue for the study of diseases such as cervical cancer or infertility. In addition, this system could allow to quickly screen several drugs or new compounds, testing their toxicity and effects on the reproductive system. Further development of this platform could include the integration of other organs, such as the immune system and the placenta, which is fundamental for the maintenance of pregnancy, to produce a more complex model of the in vivo system. In this study, the device is fabricated using modern 3D printing techniques with biocompatible resins, but the effective material biocompatibility and toxicity, as well as optical properties, are not discussed. Because the platform allows the culture of tissue fragments, the various units contain increased volumes and dilutions of nutrients with respect to that used for analysis at the cellular-level, and this may limit the accurate recapitulation of the in vivo tissue microenvironment.

## 4. Integration of Scaffold and 3D Printed Structure

The development of OoC models of the female reproductive tract significantly grew in the last 5–10 years. Previous and currently used models of the reproductive organs are still widely used and provide optimised techniques to support for specific cell growth. Some of these techniques, combined with recent success of tissue engineering and 3D printing methods, could be adopted and integrated into the OoC models described above. 

Thanks to the availability of cells from human tissues and to previous knowledge of the physiology of these organs, several groups showed the possibility to control and maintain the different phenotypes using specific media supplemented with appropriate hormones and nutrients during different stage of culture. In order to understand the crosstalk between different cell types within the uterine tissues, traditional well plate cell culture has been updated in the past using microporous inserts and modified transwell set up. Filter inserts allow both the physical separation and paracrine relationships between the cell monolayers [34]. These models have been used to study epithelial and stromal cell communication [35], invasion mechanism of endometrial cancer cells [36] and in vitro endometrial decidualization and remodelling [37,38]. However, due to the geometry of the transwell, testing cell-cell communication and diffusion of molecules through the filter is often altered by the equilibration of the liquids in the apical and basal compartments and the culture cannot be maintained for prolonged periods.

Interestingly, the adhesion and 3D organization of the cells inside a microfluidic device significantly benefit by the introduction of adhesive proteins or ECM components. In this sense, in our opinion, techniques and methods from the field of regenerative medicine and tissue engineering could improve the performance and the functions of the reproductive OoC platforms discussed above. Materials and designs are used in reconstructive surgery [39], organ transplantation [40], study of endometrial physiology and differentiation [37,41] and development of 3D models of the reproductive organs for in vitro studies [41,42].

Biomaterials used as a three-dimensional (3D) scaffold vary from natural purified ECM components (e.g., collagen and fibronectin), mixture of ECM components (e.g., Matrigel) [37,43,44,45], to synthetic biomaterials such as polyglycolic acid (PGA) [39], polylactic acid (PLA) [46], polycaprolactone (PCL) [47] and polyethylene terephthalate (PET) [48]. Despite naturally derived biomaterials have the potential to better replicate in vivo biological conditions, synthetic polymers can be easily tailored for the construction of desired scaffold microstructures with controlled shapes and properties. Biological scaffolds to be integrated in microfluidic compartments could be obtained by the decellularization of normal organs [49,50,51,52], or by building 3D microstructured matrices in collagen [53], alginate [54,55,56,57,58,59,60,61] or fibrin [62,63].

Additional improvements could derive by the adoption of culture methods used in tissue engineering for in vivo reconstruction of female reproductive organs and genital tissues. For instance, De Filippo et al. reconstituted the vaginal tissue using vaginal epithelial and smooth muscle cells isolated from female rabbits and expanded in culture onto PGA scaffolds [39]. This approach enabled to support the cell proliferation, but also showed the formation of penetrating vasculature and the ability of the implanted tissue to produce contractile forces in response to electrical impulse. 

Scaffolds have already been successfully combined with controlled perfusion inside bioreactors to develop in vitro human vaginal epithelial cells model. Low-fluid shear conditions determined cellular reorganization and ultrastructural morphology that resemble that exhibited in human vaginal tissue in vivo. Tissue integrity, identified by formation of cell-cell tight junctions, in combination with the ability of the construct to respond to a toxic test agent, showed the potential of the system for the use in microbicide screening. Nevertheless, further studies involving different pathogens and hormonal treatments will additionally validate the model for high-throughput toxicity testing. Similarly, House et al. reported a three-dimensional porous silk scaffold for the culture of human primary cervical cells under perfusion culture. The cervix-like model was tested and characterised revealing its potential for the investigation of cervical remodelling during pregnancy [42].

In a recent study, Olalekan et al. developed a 3D model of the human endometrium by recellullarization with primary cells of an acellular scaffold able to respond to a 28-day menstrual cycle hormonal treatment [41]. Despite cell proliferation, cell viability and hormone receptor expression were directly observed and significant levels of decidualization markers (i.e., prolactin) were measured from the recellularised endometrium only after the addition of cyclic adenosine monophosphate (cAMP) to the progesterone treatment, which not reliably reproduces in vivo hormonal conditions. In addition, this system might be improved by integration of different cell types to better recreate an endometrium-mimic model.

3D printing has recently emerged as a key method for the fabrication of microengineered tissue constructs that mimic natural tissue architecture and organization by patterning of cells, ECM and vasculature. [64] 3D tissue/organ printing is a technique that employs microfabrication techniques to build 3D tissue or organ-specific microenvironment by directly depositing cells or cell aggregates [65,66]. Such microfabricated 3D constructs have been widely applied for regeneration of different tissues such as skin [67], liver [68], bone [69] and cartilage [70]. In the field of reproductive biology, a recent study by Laronda et al. described the development of a 3D printed ovary composed of a porous gelatin-based scaffold for the growth and maturation of ovarian murine follicles [71]. Different porosities of the engineered constructs were tested to investigate effects of pore geometry on follicle seeding and development. The bioprosthetic ovary seeded with ovarian follicles was then implanted in ovariectomised mice resulting in a highly vascularised construct that enabled restoration of ovarian function in vivo. Furthermore, successful live birth was achieved from mice with the implanted bioprothesis, indicating hormone restoration of follicular hormonal functions. Although the innovation and potentiality of these findings, it is worth to observe that results obtained for the successful development of mouse follicles in an artificial ovary prototype might not be translatable to human models. Thus, further studies and investigations of applicability of 3D bioprinted organs for tissue engineering/regenerative medicine approaches are needed.

Bioprinting is also promising in the design of 3D tissues or disease models that can be utilised to investigate disease and cancer pathology. Existing examples of bioprinted platforms for the study of female reproductive system cancer include a 3D co-culture model of ovarian cancer [72] and an in vitro cervical tumour model [73]. The first model is fabricated using an innovative cell printing system to pattern ovarian cancer cells and normal fibroblasts within a spatially controlled microenvironment on a Matrigel substrate, while in the second 3D printing of cervical tumor cells (Hela cells) was combined with the use of ECM-mimicking biomaterials. Optimization of bioprinting process parameters enabled the fabrication of well-defined 3D bioconstructs characterised by spatial patterning precision and good structural stability, while maintaining high cell viability. The formation of 3D acinar structures and cellular spheroid observed in the ovarian cancer model and cervical tumour model, respectively, highlight the versatility and the potential of cell printing technologies in the creation of 3D tumour models for a better understanding of cancer biology and therapeutic screening. 

Despite that cell bioprinting is characterised by scalability, cost-effectiveness and accuracy on the control of cell patterning and structural properties of the scaffold, some negative aspects might be taken into consideration. These include limitations on realizable structural shapes, material biocompatibility issues and the regulation of printing time to maintain both cell viability and structure maintenance. 

The use of bioprinting and scaffold for recreating the most appropriate structure for cells has high potential for including stem cells in the organ-on-a-chip model. The use of induced pluripotent stem cells (iPSCs) derived from patients has been discussed as a method for recreating and regenerating organs’ functions in cancer patients [10]. Different paracrine factors and cytokines can be used to induce pluripotent stem cells in vitro into the primary embryonic lineage of interest [74,75], such as germ cells and somatic cells, epithelial cells of the fallopian tubes, endometrial glands or decidualised stroma. The microfluidic environment represents a great opportunity not only to improve the differentiation methods by assuring higher control of the chemical and mechanical stimulations, but also by enabling a direct assessment of the cell differentiation to confirm that the correct phenotypic characteristics and functions are expressed by the generated iPSC.

## 5. Conclusions

This review provides a brief overview of the anatomy and physiology of the main organs of the female reproductive system, with a focus on tissues’ and cells’ phenotypic characteristics during pregnancy, menstrual cycle and in specific prevalent diseases such as endometriosis. 

Because of the complex and intricate relationship between these organs, toxicology studies in the female reproductive systems have been difficult to perform. Thus, there is the need to reproduce in vitro specific functions of this system and to study the effect of compounds and toxic agents that can affect cell and organ behaviour.

The examined models represent innovative tools for the in vitro study of organs’ functions in specific conditions (e.g., ovulation, preimplantation, pregnancy) which have not been available before. The use of small amount of cells, from minimally invasive endometrial biopsies or from otherwise discarded placental tissues, allows to isolate and reprogram each single cell type and use it as a building block for the recreation of the organ in vitro. These systems enable and change significantly the approach for assessing effects of environmental insults, such as infection or inflammation, on women’s fertility and health. Once a comprehensive validation of these platforms will be completed, these models can drastically change the methods in drug testing and toxicity assessment, improving our understanding of the human physiology and replacing use of small and big animals in research with high throughput and chip assays. 

By coupling these organs with other important organs of the human body, it will possible to overcome limits of in vitro drug testing and to avoid misconception and bias generated by single gender experimental design.

These studies show that cell-cell communication has a critical role for cell growth and tissue development, and in vitro recapitulation of this mechanism is a fundamental step in the development of a microengineered device that mimics the natural in vivo structure and functions of whole organs. 

Further research is required to confirm the efficacy of these models, to identify the best sources for cells and to measure the accuracy of the results and their variability depending on the tissue’s and patients’ characteristics

## Figures and Tables

**Figure 1 bioengineering-06-00103-f001:**
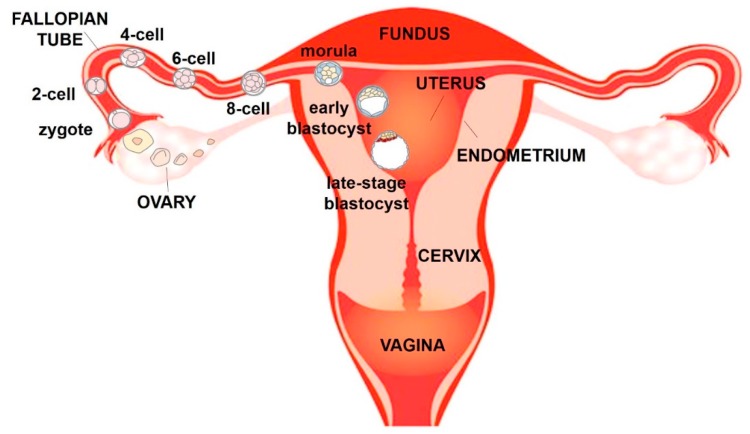
Schematic frontal view of the human female reproductive system.

**Figure 2 bioengineering-06-00103-f002:**
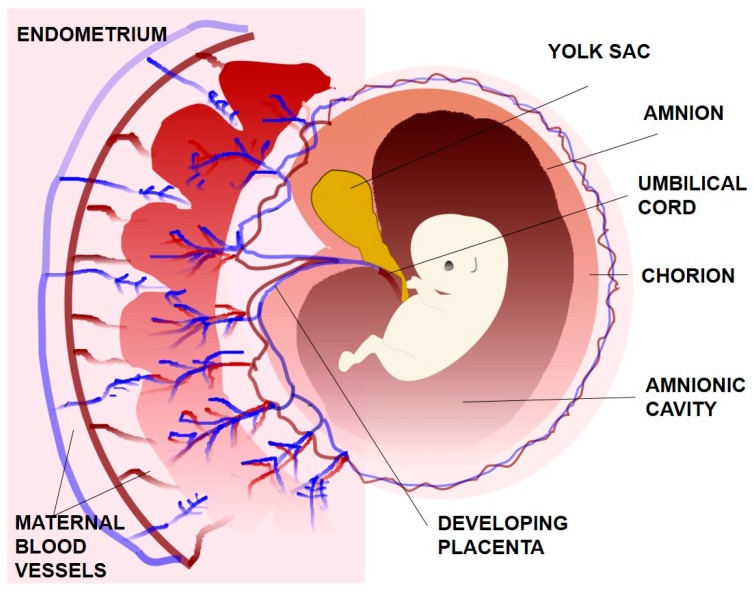
The extraembryonic membranes.

**Table 1 bioengineering-06-00103-t001:** Examples of Organs-on-chip models of the female reproductive system.

Organ Model		Placenta-On-Chip (15)	Placenta-On-Chip (16)	Uterus-On-Chip (17)	Endometrium-On-Chip (32)
**Co-cultured Cell types**		Human trophoblast (BeWo b30) cell line	Human trophoblast (JEG-3) cell line	Mouse oocyte	Human primary endometrial stromal cells
		Human primary placental villous endothelial cells	Human umbilical vein endothelial cells (HUVECs)	Mouse endometrial epithelial cells	HUVECs
**Culture Environment**	Medium	Dulbecco’s Modified Eagle Medium (DMEM) and Endothelial cell growth medium (EGM™-2)	DMEM and EGM™-2	EmbryoMax^®^ Modified M16 Medium	EGM™-2
	Temperature	37 °C	37 °C	37 °C	37 °C
	O_2_ tension	Not applicable (N.A.)	N.A.	N.A.	95%
	CO_2_	N.A.	5%	5%	5%
**Device characteristics**	Top Layer	Channel: 1 mm × 1.5 cm × 135 μm (w × l × h)	Channel: 500 μm × 200 μm (w × h)	Zigzag Channel: 500 μm × 110 μm (w × h)	Chamber: 7.75 mm × 6.2 mm
	Bottom Layer	Channel: 1 mm × 1.5 cm ×135 μm (w × l × h)	Channel: 500 μm × 200 μm (w × h)	Channels: 6 mm × 3 mm × 0.11 mm (l × w × h)	Chamber: 7.75 mm × 6.2 mm
	Device Material	Polydimethylsiloxane (PDMS)	PDMS	PDMS	PDMS
	Membrane Material	Polycarbonate (PC)	Vitrified collagen	PC	Epoxy resin (EPON 1002F)
	Membrane Pore size	1 μm	N.A.	8 μm	2 μm
**Culture characteristics**	Coating	Fibronectin	Fibronectin and gelatin	Gelatin	Matrigel
	Static versus dynamic	Dynamic	Dynamic	Dynamic	Static/dynamic
	Flow rate	100 μL h^−1^	30 μL h^−1^	10 μL h^−1^	1 μL min^−1^

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
