# Peer review of "Organs-On-Chip Models of the Female Reproductive System"

_bioengineering, 2019, doi:10.3390/bioengineering6040103_

Round 1

Reviewer 1 Report

The manuscript by Mancini and Pensabene gives a comprehensive insight in the organ-on-chip platforms applied to the female reproductive system. Few suggestions for further improving the manuscript are given below.

- The latest developments of iPSC technology push the field of organ-on-chip towards personalized medicine approaches. The use of stem cells from patients with specific diseases is become indeed rather popular to replicate pathological mechanisms in vitro. This point has been briefly mentioned in the second paragraph of section 4. The authors might expand this point and highlight new possibilities that iPSC technology can introduce in the study of the female reproductive system in vitro.

- As reported by the authors, a number of proof-of-concept organ-on-chip systems have been proposed so far in this research field, but very few have targeted pathological conditions in detail. The authors could comment on this, suggesting which pathological conditions in this area might benefit most from the use of organ-on-chip systems in the near future [this is reported in the Introduction in broad terms and might be resumed in the Conclusion].

- In the PDF file, Figure 2 seems to be missing some text labels for describing amnion and allantois (not clear). Please, revise it.

Author Response

Answers to comments of Reviewer #1:

The manuscript by Mancini and Pensabene gives a comprehensive insight in the organ-on-chip platforms applied to the female reproductive system. Few suggestions for further improving the manuscript are given below.

- The latest developments of iPSC technology push the field of organ-on-chip towards personalized medicine approaches. The use of stem cells from patients with specific diseases is become indeed rather popular to replicate pathological mechanisms in vitro. This point has been briefly mentioned in the second paragraph of section 4. The authors might expand this point and highlight new possibilities that iPSC technology can introduce in the study of the female reproductive system in vitro.

Answer:

We thank the Reviewer for the suggestion. We explored the literature and analyzed a couple of source of iPSC type that could be used for developing female organs-on-a-chip. Additional lines on this topic have been added in Section 4.

- As reported by the authors, a number of proof-of-concept organ-on-chip systems have been proposed so far in this research field, but very few have targeted pathological conditions in detail. The authors could comment on this, suggesting which pathological conditions in this area might benefit most from the use of organ-on-chip systems in the near future [this is reported in the Introduction in broad terms and might be resumed in the Conclusion].

Answer:

We agree with this observation. We included endometriosis, since many authors considered the potential of using an organ on a chip model to better understand the pathogenesis and pathophysiology of the disease. We expanded the topic at the end of each paragraph for the specific model and in the conclusions as suggested.

- In the PDF file, Figure 2 seems to be missing some text labels for describing amnion and allantois (not clear). Please, revise it.

Answer:

We corrected the figure and included the labels.

Reviewer 2 Report

This review presents an overview of recent research progress on the female reproductive system. They described a brief overview of the anatomy and physiology of the female reproductive system and discussed two topics: organ-on-a-chip models and 3D printing techniques. The manuscript is well written. However, there are some issues the authors should consider before the publication.

1) In 3.1 Placenta-on-a-chip models section, the author referred the compartmentalized microfluidic system which provide co-cultured and dynamic flow conditions. Although the authors described the benefit of the co-cultured condition, there was no clear description about how the dynamic flow condition contributed to the placenta-on-a-chip system. The author should discuss the role of the shear stress in the formation of microvilli. I recommend the citation of the paper (Shigenori Miura, et.al., Nat. Commun., 6:8871, 2015, DOI: 10.1038/ncomms9871), which uncovered the role of the shear stress in the formation of microvilli of trophoblast cells.

2) In table 1, ref 30 should be ref 29.

3) In p.9, L 316, ref 7 should be ref 8.

Author Response

Answers to comments of Reviewer #2:

This review presents an overview of recent research progress on the female reproductive system. They described a brief overview of the anatomy and physiology of the female reproductive system and discussed two topics: organ-on-a-chip models and 3D printing techniques. The manuscript is well written. However, there are some issues the authors should consider before the publication.

1) In 3.1 Placenta-on-a-chip models section, the author referred the compartmentalized microfluidic system which provide co-cultured and dynamic flow conditions. Although the authors described the benefit of the co-cultured condition, there was no clear description about how the dynamic flow condition contributed to the placenta-on-a-chip system. The author should discuss the role of the shear stress in the formation of microvilli. I recommend the citation of the paper (Shigenori Miura, et.al., Nat. Commun., 6:8871, 2015, DOI: 10.1038/ncomms9871), which uncovered the role of the shear stress in the formation of microvilli of trophoblast cells.

Answer:

We thank the Reviewer and we expanded the discussion about the dynamic perfusion, discussing the data published in the manuscript by Shinegori Miura.

2) In table 1, ref 30 should be ref 29.

3) In p.9, L 316, ref 7 should be ref 8.

Answer:

We thank the reviewer and we corrected the reference numbering.
